# Multivariate Data Analysis and Central Composite Design-Oriented Optimization of Solid Carriers for Formulation of Curcumin-Loaded Solid SNEDDS: Dissolution and Bioavailability Assessment

**DOI:** 10.3390/pharmaceutics14112395

**Published:** 2022-11-06

**Authors:** Leander Corrie, Jaskiran Kaur, Ankit Awasthi, Sukriti Vishwas, Monica Gulati, Sumant Saini, Bimlesh Kumar, Narendra Kumar Pandey, Gaurav Gupta, Harish Dureja, Dinesh Kumar Chellapan, Kamal Dua, Devesh Tewari, Sachin Kumar Singh

**Affiliations:** 1School of Pharmaceutical Sciences, Lovely Professional University, Phagwara 144411, India; 2School of Pharmacy, Suresh Gyan Vihar University, Mahal Road, Jagatpura, Jaipur 302017, India; 3Department of Pharmacology, Saveetha Dental College, Saveetha Institute of Medical and Technical Sciences, Saveetha University, Chennai 602105, India; 4Uttaranchal Institute of Pharmaceutical Sciences, Uttaranchal University, Dehradun 248007, India; 5Department of Pharmaceutical Sciences, Maharshi Dayanand University, Rohtak 124001, India; 6Department of Life Sciences, School of Pharmacy, International Medical University, Bukit Jalil, Kuala Lumpur 57000, Malaysia; 7Faculty of Health, Australian Research Centre in Complementary and Integrative Medicine, University of Technology Sydney, Ultimo, NSW 2007, Australia; 8Descipline of Pharmacy, Graduate School of Health, University of Technology Sydney, Ultimo, NSW 2007, Australia; 9Department of Pharmacognosy and Phytochemistry, School of Pharmaceutical Sciences, Delhi Pharmaceutical Sciences and Research University, New Delhi 110017, India

**Keywords:** S-SNEDDS, isomalt, Galen IQ 981, multivariate data analysis, principal component analysis

## Abstract

The study was initiated with two major purposes: investigating the role of isomalt (GIQ9) as a pharmaceutical carrier for solid self-nanoemulsifying drug delivery systems (S-SNEDDSs) and improving the oral bioavailability of lipophilic curcumin (CUN). GIQ9 has never been explored for solidification of liquid lipid-based nanoparticles such as a liquid isotropic mixture of a SNEDDS containing oil, surfactant and co-surfactant. The suitability of GIQ9 as a carrier was assessed by calculating the loading factor, flow and micromeritic properties. The S-SNEDDSs were prepared by surface adsorption technique. The formulation variables were optimized using central composite design (CCD). The optimized S-SNEDDS was evaluated for differential scanning calorimetry (DSC), Fourier transform infrared spectroscopy (FTIR), X-ray diffraction (XRD), microscopy, dissolution and pharmacokinetic studies. The S-SNEDDS showed a particle size, zeta potential and PDI of 97 nm, −26.8 mV and 0.354, respectively. The results of DSC, XRD, FTIR and microscopic studies revealed that the isotropic mixture was adsorbed onto the solid carrier. The L-SNEDDS and S-SNEDDS showed no significant difference in drug release, indicating no change upon solidification. The optimized S-SNEDDS showed 5.1-fold and 61.7-fold enhancement in dissolution rate and oral bioavailability as compared to the naïve curcumin. The overall outcomes of the study indicated the suitability of GIQ9 as a solid carrier for SNEDDSs.

## 1. Introduction

The formulation development of plant-based bioactive flavonoids is happening at a fast pace owing to their multiple pharmacological actions [1]. These include fisetin [2], quercetin [3], xanthohumol [4], apigenin [5], catechin [6] and curcumin [7]. Among them, curcumin (CUN) is considered to be the elixir of life. Although many formulation studies have been conducted, there is still scope as well as challenges that are required to be addressed for better positioning of curcumin in the market. This is due to the fact that CUN undergoes gastrointestinal degradation and has a poor dissolution rate leading to poor bioavailability [8]. To overcome these challenges, various novel formulation strategies such as liposomes [9], cubosomes [10], solid lipid nanocarriers [11], micelles [12], polymeric nanoparticles [13], dendrimers [14] and self-nanoemulsifying drug delivery systems (SNEDDSs) [15] have been adopted to incorporate these moieties and increase their solubility. SNEDDS increases the solubility of the drug after the drug is dissolved in the isotropic mixture containing oil, surfactant and co-surfactant. When taken orally, SNEDDS generates oil-in-water nanoemulsion with droplets size lesser than 200 nm [16]. The benefit of these tiny droplets of nanoemulsion is that they deliver the drug in soluble form with a high interfacial surface area for drug absorption, resulting in improved, more consistent and repeatable bioavailability as well as improving the penetration of the drugs from gastrointestinal mucosa. SNEDDS as a drug delivery system provides added benefits by protecting the active drug moiety from gastrointestinal degradation and first-pass metabolism as the drug is absorbed by the lymphatic system [17]. However, in spite of these advantages, liquid SNEDDSs (L-SNEDDSs) suffer from various issues such as stability, microbial growth, precipitation and cracking [18].

To overcome the challenges of L-SNEDDSs, solidification of L-SNEDDSs is performed using various techniques such as surface adsorption technique [19], hot melt extrusion technique [20], spray drying [21] and lyophilization [22]. Solid SNEDDSs (S-SNEDDSs) are lipid-based drug delivery systems developed by solidifying liquid excipients into powders; they offer a viable drug delivery system for poorly water-soluble molecules because they integrate the benefits of L-SNEDDSs (solubility and bioavailability enhancement) and solid dosage forms (high storage and mechanical stability, ease of administration, various dosage form options). Adsorption of L-SNEDDSs on the surface of a porous carrier is the most extensively implemented and cost-effective approach [23]. Enhanced oil loading efficiency, minimized process failure, simple formulation technique and rapid transformation of the solidified powder formulation into oral solid dosages are the major advantages of the adsorption method [24]. There are many carriers that have been used for the preparation of S-SNEDDSs, including lactose [25], d-mannitol [26], microcrystalline cellulose [27], poly vinyl alcohol (PVA) [28], *Ganoderma lucidum* powder [29] and magnesium stearate [30]. However, these carriers suffer from various challenges such as poor flow properties, bulkiness of the material and inability to compress the chosen carriers [31]. Due to the constraints of the above-mentioned carriers, a unique, cost-effective strategy has been adopted to incorporate isomalt (Galen IQ) as a carrier for SNEDDSs for the first time. Based on the properties such as excellent flowability, high physical stability upon mixing, high dilution potential, specific morphology preventing segregation, suitability for all patients and low hygroscopicity, it was chosen as a carrier for formulating S-SNEDDSs [32].

In the present study, an attempt has been made to develop CUN-loaded S-SNEDDSs owing to the advantages of S-SNEDDSs in overcoming dissolution rate and gastrointestinal degradation-limited oral bioavailability of poorly soluble bioactive compounds as well as enhancing the challenges related to instability of L-SNEDDSs. Initially, the impact of various solid carriers was investigated on precompression properties of S-SNEDDSs to select the best carriers and coating materials for developing S-SNEDDSs through preliminary screening using multivariate data analysis. Afterward, the quality by design (QbD) approach was used to investigate the effect of selected carriers on various responses, viz. angle of repose, Hausner’s ratio and entrapment efficiency. The advantages of QbD include identifying the variability of process parameters and material attributes, decreasing batch failure, increasing method robustness and ensuring the method achieves the intended performance. Moreover, the efficiency of developed CUN-S-SNEDDSs was evaluated based on enhancement in dissolution rate, area under curve (AUC) and oral bioavailability of CUN. Furthermore, the optimized formulation was characterized by differential scanning calorimetry (DSC), X-ray diffraction studies (XRD), scanning electron microscopy (SEM) and pharmacokinetic studies, which are discussed in subsequent sections.

## 2. Materials and Methods

### 2.1. Materials and Equipments

CUN was purchased from M/S Himedia, Mumbai, India. Propylene glycol (PG), Tween (T) 20 and 80; Polyethylene glycol (PEG) 200, 400 and 600: Span 20 and 80, Aerosil 200 (AER-200): hydroxy propyl methyl cellulose (HPMC); and sodium carboxy methyl cellulose (NaCMC) were purchased from Loba Chemie, Mumbai, India. Labrasol, Transcutol P (TRP) and Labrafil M1944 CS (LAB) were obtained as gift samples from M/s Gattefosse in Mumbai, India. Syloid XDP 3514 (SXDP) was obtained as a gift sample from Grace Materials Technologies, Discovery Science, Pune, India. Galen IQ 981 (GIQ9) and Galen IQ 721 (GIQ7) were obtained as gift samples from SFA Food and Pharma Ingredients, Thane, India. All other chemicals used were of analytical grade. In order the quantify the amount of drug, RP-HPLC was performed (SPDM20A; Shimadzu, Kyoto, Japan) at 420 nm, all experiments were carried out in triplicate and mean data were recorded.

### 2.2. Methods

#### 2.2.1. Formulation of L-SNEDDSs

Initially, studies on the solubility of CUN in various oils and surfactants were carried out. The results of solubility studies indicated that CUN had the highest solubility in LAB, T80 and TRP. Therefore, the isotropic mixture consisting of these solubilizers was selected for the preparation of L-SNEDDSs of CUN. The L-SNEDDS prototype was prepared by mixing LAB (100 µL), T80 (450 µL) and TRP (450 µL) to give 1 mL of isotropic mixture, which was vortexed with 25 mg of CUN for 10 min. The drug preconcentrate (1 mL) was added into a beaker containing 500 mL of water that was rotated on a magnetic stirrer at 500 rpm maintained at 37 ± 0.5 °C for 5 min [22].

#### 2.2.2. Solidification of L-SNEDDSs

Various solid carriers such as GIQ9, GIQ7, SXDP, AER-200, HPMC and Na CMC were used for the solidification of L-SNEDDSs. The solidification was performed by surface adsorption technique.

#### 2.2.3. Determination of Loading Factor

To achieve better flow and compaction properties, it was important to know the loading factor of the various carriers. The loading factor (Lf) was determined by measuring the amount required for transforming the unit dose of liquid oily formulation into the solid free-flowing powder [33]. The Lf is given by Equation (1). The study was carried out in triplicate and mean data were recorded.
(1)$$\mathrm{Lf}=\frac{\mathrm{Amount}\;\mathrm{of}\;\mathrm{liquid}\;\mathrm{isotropic}\;\mathrm{mixture}}{\mathrm{Amount}\;\mathrm{of}\;\mathrm{carrier}\;\mathrm{material}}$$

#### 2.2.4. Screening of Solid Carriers

The prepared formulations using various carriers were subjected to flow property analysis. The study characterized the use of various solid carriers for angle of repose (AR), bulk density (BD), tapped density (TAD), true density (TRD), compression index (CI) and Hausner’s ratio (HR). The study was carried out in concordance with the procedure mentioned by Khursheed et al. [29]. AR was evaluated by fixed funnel and free-standing cone method. Briefly, a known weight of powder was poured through a funnel until the apex of the formed cone touched the tip of the funnel [34]. The radius of the cone and its height were determined, and the AR was calculated using the given Equation (2), where “h” is the height of the cone and “r” is the radius of the powder formed.
(2)$$\left(\mathrm{AR}\right)ϴ=\tan^{-1}\left(\frac{h}{r}\right)$$

The BD was determined using a measuring cylinder [35]. Briefly, a known amount of powder was poured into the measuring cylinder at a slant position, the initial bulk volume was noted and the bulk density was determined using Equation (3). Similar to BD, TAD was calculated after 100 taps and was determined using Equation (4). TRD was measured using the benzene displacement method. Here, W1 = weight of the powder, SG = specific gravity of the solvent, W2 = weight of the bottle and solvent and W3 = weight of bottle along with solvent and powder and was calculated as shown in Equation (5). The CI and HR were calculated using Equations (6) and (7). The study was carried out in triplicate and mean data were recorded.
(3)$$\mathrm{BD}=\frac{\mathrm{Weight}\;\mathrm{of}\;\mathrm{powder}}{\mathrm{Bulk}\;\mathrm{Volume}}$$
(4)$$\mathrm{TAD}=\frac{\mathrm{weight}\;\mathrm{of}\;\mathrm{powder}}{\mathrm{Final}\;\mathrm{volume}\;\mathrm{after}\;\mathrm{taps}}$$
(5)$$\mathrm{TRD}=\left(\frac{W1}{\left(W2+W1\right)-W3}\right)\times \mathrm{SG}$$
(6)$$\mathrm{CI}=\frac{\mathrm{Tapped}\;\mathrm{Density}-\mathrm{Bulk}\;\mathrm{Density}}{\mathrm{Tapped}\;\mathrm{Density}}\times 100$$
(7)$$\mathrm{HR}=\frac{\mathrm{True}\;\mathrm{Density}}{\mathrm{Bulk}\;\mathrm{Density}}$$

#### 2.2.5. Kawakita Plots for Solid Carriers

A standard procedure was used to determine the flowability of powders for the estimation of Kawakita analysis. Briefly, a known quantity of powder was poured into a 50 mL measuring cylinder and the bulk volume was noted [36]. Tapping was performed for the measuring cylinder, and the volume was noted after fixed number of taps (N). Kawakita analysis was performed and analyzed using Equation (8). Here, “a” and “b” are constants. The volume of shrinkage due to tapping is indicated by the letter “a”, which is also known as compactibility. Cohesiveness is defined as “1/b”, and the decrease in the degree of volume is indicated by “C”. The calculation was performed using the initial volume (V0) and tapped volume (VN) as inputs, as shown in Equation (9). Based on the slope of the graph involving “N/C” and the number of taps “N” (i.e., N = 5, 10, 15, 20 and 25), the readings were noted for the constants “a” and “1/b”.
(8)$$\frac{N}{C}=\frac{N}{a}+\frac{1}{\mathrm{ab}}$$
(9)$$C=\frac{V_{0}-{\;V}_{N}}{V_{0}}$$

#### 2.2.6. Multivariate Data Analysis

Understanding the interrelationships between variables in complicated datasets was achieved through multivariate data analysis. Classifying the process variable in accordance with the impact of each particular characteristic may be useful. Principal component analysis (PCA) is a multivariate analysis technique that linearly reduces a large number of variables to a smaller number of well-chosen key variables. Multiple variables are reduced in number and scope using PCA’s concept of dimensionality reduction. The purpose of PCA in this study was to understand the different qualities of powders used and to give a theoretical justification for screening suitable carriers for the development of CUN-S-SNEDDSs [37]. The PCA plots were drawn using the Unscrambler software.

#### 2.2.7. Design of Experiment (DoE)

DoE tools aid in identifying and explaining how important critical process parameters (CPPs) (independent variables) influence critical material attributes (CMAs) (dependent variables), enhancing the formulation targeting profile [38]. From an industrial standpoint, DoE plays an important role in considerations related to technology transfer and scale-up [39]. From the screening of carriers and coating ratios as well as multivariate data analysis, it was understood that GIQ9 and AER-200 possessed the desired qualities required for the formulation of CUN-S-SNEDDSs such as enhanced flow properties and excellent loading factor. Hence, these were used as carriers for the development of the CUN-S-SNEDDSs. Based on the Lf values, 80%, 100% and 120% of the carrier Lf values were selected as the levels for the central composite design (CCD). The design generated a total of 14 runs for converting the L-SNEDDS to a CUN-S-SNEDDS using the surface adsorption technique. The L-SNEDDS formulation was introduced to the inert solid carriers, GIQ9 and AER-200. The carriers in the ratio of 1:1 were added to the L-SNEDDS and blended in a blender for 5 min to obtain an even blend and then left for drying. Then, AOR, HR and entrapment efficiency (EE) were characterized based on previously described sections, and their mean and standard deviation were recorded.

#### 2.2.8. Entrapment Efficiency (EE)

The entrapment efficiency percentage of all the formulations mentioned in DoE was calculated. Briefly, the SNEDDS loaded with CUN was added to 500 mL of water and then stirred for 5 min [40]. Following this, 2 mL of sample was removed and centrifuged at 10,000× *g*. The supernatant was removed and passed through a 0.45 µm filter and analyzed using an HPLC instrument at 420 nm. The entrapment efficiency was calculated by the formula given in Equation (10).
(10)$$\mathrm{Entrapment}\;\mathrm{Efficiency}\;\left(\%\right)=\frac{\mathrm{Amount}\;\mathrm{of}\;\mathrm{drug}\;\mathrm{in}\;\mathrm{formulation}}{\mathrm{Amount}\;\mathrm{of}\;\mathrm{drug}\;\mathrm{added}\;}\times 100$$

#### 2.2.9. Droplet Size and Zeta Potential

Dynamic light scattering (Zetasizer 2000 HS, M/s Malvern Instruments Limited, Malvern, UK) was used to ascertain globule size and zeta potential at room temperature. All assessments were performed with a detection angle of 90° and a laser light with a potential of 50 mV [41]. Prior to the analysis, the SNEDDS formulations were passed through a 0.2 µm filter and transferred to polystyrene cuvettes for analysis.

#### 2.2.10. Fourier Transformed Infrared Spectroscopy (FTIR)

To ascertain the interplay among excipients and the active drug, as well as to understand the functionality of the groups of CUN, GIQ9, AER-200 and CUN-S-SNEDDS, FTIR was performed. FTIR was carried out in the wavenumber range of 400–4500 cm^−1^ and the transmittance percentage was recorded [42].

#### 2.2.11. Differential Scanning Calorimetry (DSC) Analysis

DSC 6000 (PerkinElmer, Waltham, MA, USA) was used to document the DSC pattern of naïve CUN, GIQ9, AER-200 and CUN-S-SNEDDS. To assess the stability and functionality, the samples were placed in an aluminum pan and were analyzed at a temperature increase of 10 °C/min across a temperature gradient of 20–300 °C in an inert nitrogen atmosphere at a flow rate of 50 mL/min [43].

#### 2.2.12. Powder X-ray Diffraction Analysis (PXRD)

The PXRD pattern of naïve CUN, GIQ9, AER-200 and CUN-S-SNEDDS was recorded using a diffractometer system (Bruker axs, D8 Advance, Billerica, MA, USA) with copper Kα radiation. The voltage and current of the tube were set to 45 kV and 40 mA, respectively [44]. Each sample was packed in an aluminum sample container and evaluated at ambient room temperature using a continuous scan between 10 and 80° at a scan speed of 0.010° min^−1^.

#### 2.2.13. Field Emission Scanning Electron Microscopy (FESEM)

The surface morphology of CUN, GIQ9, AER-200 and CUN-S-SNEDDS was visualized using FESEM studies. Prior to evaluation, specimens were positioned on aluminum stubs with double-sided adhesive tape and sputter coated with a thin layer of gold at 10 Torr vacuum. The specimens were scanned with an electron beam at 10 kV acceleration potential, and images were captured in secondary electron configuration with the FESEM device [45].

#### 2.2.14. Transmission Electron Microscopy (TEM)

A drop of vesicle suspension was dropped and adsorbed onto a copper grid covered in carbon film. Subsequently, a 1% (*w*/*v*) solution of phosphotungstic acid (PTA) was used to negatively stain the copper grids. The dyed grids containing the samples (CUN, GIQ9, AER-200 and CUN-S-SNEDDS) were then dried and examined by TEM (Hitachi H-7000). The instrument had a digital camera and operated at a voltage of 200 kV (Mega View II, Olympus, Tokyo, Japan) [46].

#### 2.2.15. In Vitro Dissolution Studies

To understand the effect of various formulation variables on drug release, dissolution studies were performed. The USP type I dissolution apparatus was employed. This was rotated at a constant speed of 100 ± 5 rpm and maintained at a temperature of 37 ± 0.5°C [47]. An amount equivalent to 25 mg of CUN was subjected to dissolution for naïve CUN, CUN-L-SNEDDS and CUN-S-SNEDDS. In size “000” capsules, the aforementioned contents were added individually. The study was carried out by mimicking the gastric fluid in simulated gastric fluid (SGF) (pH 1.2). At predefined sample time points of 5, 10, 15, 30, 45 and 60 min, the samples were removed and centrifuged at 10,000× *g* for 5 min. The supernatant was withdrawn and injected into the HPLC instrument, and the drug release was measured at 420 nm.

#### 2.2.16. In Vivo Studies

##### Procurement and Ethical Clearance

Wistar rats were purchased from the Lala Lajpat Rai University of Veterinary and Animal Sciences (LUVAS), Hisar, India. For the study, 8–10-week-old rats with a body weight of 220–250 g were selected. Prior to the treatments, they were enclosed in polypropylene animal cages lined with husk and acclimatized for a week to a 12 h day-and-night cycle at a temperature of 27 ± 3 °C and relative humidity of 50 ± 5%. The rats were fed a standard diet and given unlimited access to water [48]. The protocol was approved by the institutional animal ethical committee of Lovely Professional University, bearing the number LPU/IAEC/2021/86.

##### Bioanalytical Method Development

A bioanalytical method for the estimation of CUN in biological plasma samples was carried out according to the ICH M10 guidelines. A C18 column was used as the stationary phase, and methanol and water were used as mobile phases in the isocratic method. The instrument HPLC LC-20AD, Shimadzu, Japan, was used. Quercetin was used as an internal standard. The run time was 20 min. Quercetin and CUN had retention times of 4.4 and 14.18 min, respectively [49].

##### Collection of Blood and Plasma

Blood samples of 0.75 mL were drawn from the retro-orbital plexus of each group of rats at predetermined time intervals. They were allowed to flow through a capillary into RIA vials containing ethylene diamine tetra acetic acid (EDTA) crystals and then centrifuged at 3000 *g* for 15 min. The plasma was collected and stored at −40 °C until further evaluation [49].

##### Pharmacokinetic Study

For in vivo pharmacokinetic studies, a single-dose parallel-design study was carried out. The animals were divided into two groups, each with six animals. Group 1 received naïve CUN at a dose of 25 mg/kg suspended in 0.5% CMC *w*/*v*, orally. Group 2 received CUN-S-SNEDDS, orally at a dose equivalent to 25 mg/kg. The formulations were suspended in 1 mL of 0.5% CMC solution. The blood (0.3 mL) was collected in EDTA vials and centrifuged at 3000× *g* for 15 min. The plasma sample was collected for further processing. The processed samples were chromatographically analyzed using an initially developed and validated HPLC method for CUN in rat plasma. For pharmacokinetic analysis of plasma drug concentration, PKSolver 2.0 software was used. Pharmacokinetic parameters such as C_max_, AUC, C_max_/AUC, T_max_, Ka and K were calculated and statistically validated [50].

##### In Vitro/In Vivo Correlations (IVIVCs)

Linear and nonlinear point-to-point level A IVIVCs were sought between in vitro percent drug released and in vivo percent drug absorbed at the correlating time for naïve CUN and CUN-S-SNEDDS, and the statistically significant difference of every correlation was determined based on the respective F-ratio values [51]. Since the drug was observed to have one-compartment open-body model pharmacokinetics, the values of cumulative percent drug absorbed at specific time points were computed using a modified Wagner–Nelson method [52].

#### 2.2.17. Statistical Analysis

The mean and standard deviation (SD) were used to express all experimental data. Utilizing GraphPad Prism version 7.0 (GraphPad Software Inc., San Diego, CA, USA), statistical examination of the collected data was performed using either the analysis of variance or Tukey’s multiple comparison test. The results were considered significantly different when the value of *p* < 0.05 was obtained.

## 3. Results and Discussion

### 3.1. Screening of the Carrier and Coating Material

All the carrier and coating materials were subjected to micromeritic evaluation and Lf determination. The Lf and oil adsorption capacity values for GIQ9, GIQ7, SXDP, AER-200, HPMC and NaCMC were found to be 0.57 (1.75 g), 0.63 (1.58 g), 1.9 (0.52 g), 2.1 (0.476 g), 0.67 (1.49 g) and 0.59 (1.69 g), respectively. It was inferred from Lf values that a higher Lf value corresponded to a lesser amount of carrier required to convert the oily formulation into a free-flowing powder. The Lf values for AER-200 and SXDP were found to be the highest in this case. There was a great degree of variation in the case of AR exhibited by these carriers. The ARs for all the carriers were found to be in the range of 23.51° and 50.53° for SXDP and NaCMC, respectively. Similarly, the BD, TAD and TRD were found to be highest for GIQ9, 0.66 g/cm^3^, 0.76 g/cm^3^ and 0.71 g/cm^3^, respectively, and the lowest for AER-200, 0.04 g/cm^3^, 0.06 g/cm^3^ and 0.05 g/cm^3^, respectively. This was attributed to the change in the physicochemical properties of the carrier systems used. From the Kawakita plots, the compactibility and cohesion of the carriers were noted. The Kawakita equation describes the characteristics related to the volume reduction of powder and the pressure that is applied for a powder placed in a confined vessel. Linear graphs were obtained from the Kawakita graphs [53]. The compactibility “a” and cohesiveness “1/b” derived from these are shown in Table 1, indicating the compactibility and consolidation properties of the respective carriers. The compactibility value of GIQ9 was 0.099, which was higher than the compactibility values of other carriers. This revealed that the higher the compactibility, the lower the cohesiveness of the particles. It was noticed that SXDP had better fluidity, whereas GIQ9 had better compactibility. Hence better the propensity of GIQ9 was confirmed. The HR was found to be lowest for GIQ9 (1.15) and highest for NaCMC (2.05). The alterations in the flow properties of the powders were attributed to differences in the physicochemical and physicomechanical properties of the powders. The precompression parameters such as Lf, oil adsorption capacity, BD, TAD and TRD are very important for assessing the flow behavior of a solid formulation as they significantly impact the flow and compression properties. They ultimately impact the friability, hardness, disintegration and dissolution rate. In addition, when a liquisolid formulation is handled, the AR and flow rate significantly impact the propulsion properties, storage, stability and powder agglomeration [35].

### 3.2. Multivariate Data Analysis

Understanding the interrelationships among variables in complicated datasets has been made possible through multivariate data analysis. Arranging the process variable by the respective property’s influence decreases the chance of variability [54]. The PC1 plot versus the PC2 plot shown in Figure 1a shows the relationship between the carriers used and their measured physicomechanical properties. The model demonstrated a variance of 98% (88% for PC1 and 10% for PC2). Additionally, it is possible to notice that runs with similar features were grouped, whilst runs with significantly dissimilar qualities were situated on opposing ends. The score plot demonstrated that the most significant variable should have high principal component covariance and correlation. The biplots helped determine the correlation between variables and the responses [55]. It was noticed from the biplot that GIQ9 was strongly anticorrelated with the loading factor and cohesion of the particles, whereas NaCMC showed better cohesiveness and improper loading factor. It can also be inferred from the biplot that GIQ7, HPMC and NaCMC were the runs that were grouped together at the center of the plot. It was recognized that all of the characteristics with highly positive or negative PC loading values were associated with the powder’s physical properties. It was also inferred that the loading factor was negatively correlated with the cohesion of the powder. Additionally, the main variables affecting the carriers’ variability were examined using a partial least square (PLS) model to ascertain how they relate to one another. The model developed for the loading factor was successfully validated using the PLS model, shown in Figure 1b [56]. The red dots indicate the validation dataset, whilst the blue dots indicate the calibration. All experimental runs revealed a significant association between predicted and observed loading factors. The corresponding R^2^ values for predicted and observed loading factors were 0.9921 and 0.9496, respectively. It was noticed that there was no variability between the R^2^ values of predicted and obtained loading factors. Hence, it was understood that the model can be used to adequately define the loading factor of the powders.

### 3.3. Design of Experiment (DoE)

The impact of important factors on the AR, HR and EE of the CUN-S-SNEDDS was investigated using the CCD. For easier computations and to enable the findings to be orthogonal, the real values of the independent variables were converted. All the response values of the factors are given in Table 2.

AR, HR and EE were found to be in the ranges of 31–38.4°, 1.12–1.25 and 89.3–98.2%, respectively, as a result of combinations of various values of independent variables. Linear, two-factor interaction (2FI) and quadratic models were used to fit the results. The best-fit models for each response were chosen based on the correlation coefficient (R^2^), projected R^2^, adjusted R^2^ and standard deviation (SD). The ratios of maximum to minimum for the responses AR, HR and EE were 1.23, 1.11 and 1.17, respectively. Since the ratio of maximum to minimum was less than 10, the power of transformation was not required. The quadratic model was found to be the best-fitted model for all the responses. Coefficients with only one term, a combination term and higher-level terms represent the primary interactions and quadratic effects. To ascertain the statistical relevance and degree of each variable’s main effects as well as their interactions, analysis of variance (ANOVA) was used. The contour plots shown in Figure 2 for independent variables were created using the generated regression model. The model’s suitability was confirmed by the ANOVA table (*p* < 0.05), which also reveals the important variables that have an impact on the development of the CUN-S-SNEDDS responses Y1, Y2 and Y3. The concentration of GIQ9 was found to be a key model term for AR (Y1). Although the effect was very small, the concentration of AER-200 affected the AR. However, both the concentration of GIQ9 and the concentration of AER-200 had a substantial impact on the HR and EE. In terms of coded factors, the mathematical equations for the responses were determined and are given in Equations (11)–(13).
(11)$$\mathrm{Angle}\;\mathrm{of}\;\mathrm{repose}\;\left(Y1\right)=+36.95-2.14A-0.4993B+0.9500\mathrm{AB}-1.29A^{2}-1.49B^{2}$$
(12)$$\mathrm{Hausner}\;\mathrm{ratio}\;\left(Y2\right)=+1.22-0.0370A+0.0018B-0.0050\mathrm{AB}-0.0016A^{2}-0.0411B^{2}$$
(13)$$\mathrm{Entrapment}\;\mathrm{Efficiency}\;\left(Y3\right)=+95.37+3.91A-0.2413B+0.2750\mathrm{AB}-2.25A^{2}-0.0708B^{2}$$

A positive sign denotes an additive impact (synergistic effect), while a negative sign denotes a diametrically opposed effect (antagonistic effect). In the case of AR (Y1), a negative coefficient of factor A represented a decrease in AR with an increase in the concentration of GIQ9. This effect was similar for factor B.

However, in the case of HR, factor A showed an antagonistic effect and factor B showed a synergistic effect. In the case of entrapment efficiency, factor A showed a synergistic effect and factor B showed an antagonistic effect. Despite showing an additive impact or opposed effect, the results of the effects may not be statistically significant. This can be better understood by perturbation plots that demonstrate the factors which affected the responses by virtue of a steep curve, shown in Figure 1. It was noted that the concentration of GIQ9 was the factor that most influenced AR. In the case of HR, the concentration of AER-200 played an important role. However, in the case of EE, the concentration of GIQ9 played an important role, suggesting its adsorption capabilities and demonstrating its role as a solid carrier for poorly soluble drugs. The polynomial equations were generated to better understand the developed design. It was noticed in the case of AR, as the concentration of GIQ9 and AER-200 increased, the AR reduced. In the case of HR, as the concentration of GIQ9 increased, the HR reduced. For AER-200, the contour plot showed that an increase in the concentration of AER-200 caused an increase in HR up to a certain extent and then a decrease. In the case of EE, the contour plot revealed that as the concentration of GIQ9 increased, EE increased. However, EE slightly decreased as the concentration of AER-200 increased. In summary, it was noticed that in the solidification of L-SNEDDSs, the role of GIQ9 was more dominating as compared to AER-200 for AR and EE. This was attributed to the fact that GIQ9 demonstrated better cohesion, better compactibility, better adsorbability and better flow rate in spite of having a lower Lf value. These properties became much better upon the addition of AER-200. However, when the consideration of unit dosage forms is taken into account, AER-200 may be used in combination with GIQ9 which may aid in reducing the size and quantity required for the excipients in the preparation of oral solid dosage forms, thereby enhancing the overall effect of GIQ9.

### 3.4. Graphical Optimization

Optimization of the developed CUN-S-SNEDDS was performed to find the levels of concentration of GIQ9 and AER-200. After plotting the desired values of dependent and independent variables on the software, it suggested that for the factors X1 and X2, the values were 2.07 and 0.38 g, and AR (Y1), HR (Y2) and EE (Y3) were 31.5°, 1.14 and 96.92%, respectively. The overlay plot, shown in Figure 3 demonstrated a yellow region which indicated the robustness of this model [38]. Moreover, it also showed a good desirability of 0.88. Using the values of the factors, CUN-S-SNEDDSs were formulated, and the AR, HR and entrapment efficiency were found to be 31.4°, 1.13 and 96.73% respectively. There was no significant difference (*p* < 0.05) between predicted and obtained values. This indicated the reproducibility of the optimized method. Since these values were found close to the predicted values, the developed CUN-S-SNEDDSs were processed for further characterization.

### 3.5. Droplet size and Zeta Potential

It was found that L-SNEDDS showed a droplet size, zeta potential and PDI value of 77.23 ± 0.56 nm, −23.42 ± 0.45 mV and 0.213, respectively. After solidification, the droplet size, zeta potential and PDI value increased to 97.54 ± 0.76 nm, −26.82 ± 0.34 mV and 0.354, respectively, for CUN-S-SNEDDS. The increase in the negative surface charge of the CUN-S-SNEDDS was attributed to the adsorption of the CUN-L-SNEDDS on the negatively charged carriers of GIQ9 and AER-200. Chemically, GIQ9 is composed of a combination of 6-O-α-d-glucopyranosyl-d-sorbitol (1,6-GPS) and 1-O-α-d glucopyranosyl-d-mannitol dihydrate (1,1-GPM) units which consist of many OH groups, giving it a negative charge [57]. Moreover, AER-200 is chemically silicon dioxide (SiO^2^), which provides a layer of negative charge facilitating Coulombic attractions [2]. Moreover, in the case of GIQ9, the negative allocation to the free energy from the breakdown of isomalt–isomalt and potential drug–drug hydrogen bonds is greater than the positive contribution from the heat of mixing, thereby giving it a negative charge [58]. Upon solidification, the droplet size was increased; this was attributed to the fact that CUN-L-SNEDDSs were adsorbed onto the solid carrier. This was also attributed to the surface adsorption technique leading to unagglomerated powder. Further, the PDI was increased to 0.345, which indicated the lesser heterogeneity and more uniformity upon solidification and usage of the solid carriers. Droplet size is an important factor, as it shows an effect on the enthalpic and entropic qualities that control the degree of adhesion between nanoparticles and cellular receptors, thereby controlling cellular uptake. Droplet size is an important parameter as it affects the physical and chemical stability of the nanoemulsion and is linked to the viscosity of the oil that was used to prepare it [59].

Lower-viscosity oils give nanoemulsions even lower in the nanometer range. In this case, LAB and TRP formed the lower-viscosity oil phase which contributed to the nanosized range of the developed SNEDDSs. The ideal situation for a given system is one that minimizes droplet size because nanodroplet systems are more transparent than conventional systems, which reduces the heterogeneity of the system and leads to higher stability [60]. Zeta potential, on the other hand, is important as it is linked to the pharmacokinetics of the formulation. A nanoparticle’s behavior in a biological context can be greatly affected by changing characteristics including size, shape and charge. Numerous studies revealed that negatively charged nanoemulsions clear more quickly and have stronger reticuloendothelial absorption than neutral or positively charged nanoemulsions [61]. In this case, since CUN-S-SNEDDSs had a higher negative charge, the developed formulation using GIQ9 can have better reticuloendothelial absorption, increasing the therapeutic activity of the drug CUN. Moreover, a higher zeta potential value corresponds to better stability and less flocculation [62]. Compared to the CUN-S-SNEDDSs developed in the current study, several of the earlier investigations on SNEDDSs conducted by other researchers either reported a higher droplet size or found the zeta potential to be lower. In the study carried out by Kazi and coworkers, it was found that the developed SNEDDS for CUN was 700 nm and had a zeta potential of −14.5 mV [63]. In another study, L-SNEDDSs of curcumin were prepared by Shukla et al. for anticancer activities; the droplet size was 83.27 nm and the zeta potential was −16 mV [64]. In the present study, the droplet size and zeta potential for CUN-L-SNEDDSs were 77.23 nm and −23.42 mV, respectively. These were far superior to the previously published results and indicated improved reticuloendothelial absorption of CUN-SNEDDSs due to the presence of GIQ9 as a carrier.

### 3.6. FTIR Analysis

As compared to other β-diketones, CUN is expected to exist in enol configuration. In the FTIR spectra, CUN showed its characteristic peaks. The appropriate properties of CUN have led to the identification of the bands at 3553 and 1427 cm^−1^ attributed to -OH stretching and -OH in-plane displacement vibrations. The clear, precise and intense bands at 1624, 1605 and 1506 cm^−1^ have been attributed to vibrations caused by carbonyl stretching. Strong and medium strong bands at 1264 and 1234 cm^−1^ have been attributed to C-O stretching vibrations, respectively. Weak bands at 2925 cm^−1^ and 2854 cm^−1^ have been attributed to the asymmetric and symmetric vibrations of the CH3 group, respectively. The C=C stretching vibration bandwidth in a conjugated system is shorter than that of an independent C=C group. A band at 1712 cm^−1^ has been ascribed to the C=C stretching vibration [22]. GIQ9 also exhibited characteristic peaks that included the vibrational spectra in the ranges of 657 to 1508 cm^−1^ and 2803 to 3608 cm^−1^ [65]. The IR spectra linked with abundant C-C-H wagging were identified due to the transitions from 1150 to 1500 cm^−1^. The characteristic -OH stretching was noticed at 3284 cm^−1^ due to the presence of 1-O-α-D-glucopyranosyl-D-mannitol dihydrate in GIQ9. The FTIR spectrum of AER-200 demonstrated a strong -OH stretching absorption band at 3234 cm^−1^, as well as a strong Si-O-Si linkage observed at 1070.24 cm^−1^ [22]. The final formulation also demonstrated a sharp peak at 1065.32 cm^−1^ attributed to the Si-O-Si bond and the -OH peak of GIQ9 at 3372.89 cm^−1^, indicating that CUN was adsorbed onto the carriers and facilitated improved solubility. The results of the FTIR study are given in Figure 4a.

### 3.7. DSC Analysis

The DSC thermogram of raw CUN exhibited a sharp endothermic peak at 173 °C which might be due to the melting point of the drug. The DSC thermogram of GIQ9 showed the presence of two endothermic peaks at 99.47 °C and 160.77 °C which were attributed to the presence of two sugar alcohols (stereoisomers), namely gluco-mannitol and gluco-sorbitol, indicating the crystalline nature of GIQ9. The first endothermic peak of GIQ9 is due to the loss of weight attributed to the dehydration of glucopyranosyl mannitol-bihydrate. This indicated the presence of numerous hydroxy groups contained in the excipient molecules and the strong linkages between the water crystallization molecules [65]. The second endothermic peak of GIQ9 is more symmetric and narrower, indicating the melting of the sugar. When heated in an oven at a gradual rate, crystallinity can be substantially restored [66]. Even though it is stable at temperatures up to roughly 250 °C when heated, after dehydration, isomalt tends to absorb humidity from the air if there is no protection of the sample. The DSC thermogram of AER-200 exhibited omission of any defined peaks, indicating its amorphous nature. The CUN-S-SNEDDS formulation showed amorphous nature by the presence of its halo pattern indicating the complete absorption of the drug on the carriers GIQ9 and AER-200. Further, the absence of crystalline peaks of the drug or GIQ9 indicated the excellent adsorption of the poorly soluble drug on such carriers. If CUN or GIQ9 had been precipitated, it would have appeared as individual peaks in the final formulation. This did not happen, and the final formulation showed amorphous characteristics. To gain a better perspective of the formulation, the DSC studies were compared to PXRD and FESEM studies. The DSC thermogram results are given in Figure 4b.

### 3.8. PXRD Analysis

The PXRD spectra of raw CUN displayed diffraction angles of 2ϴ at 4°, 12°, 16°, 17°, 18°, 24° and 26°, indicating the crystalline nature of the poorly soluble drug. However, GIQ9 showed PXRD spectra with diffraction angles of 2ϴ at 14°, 15°, 17°, 19°, 22°, 24° and 28°. This confirmed the crystalline nature of GIQ9 with the absence of any polymorphism [67]. The PXRD spectra of AER-200 and CUN-S-SNEDDS showed the absence of any peaks and the presence of a halo sequence indicating the amorphous nature of AER-200 and the final formulation. This further verified the fact that the drug isotropic mixture was adsorbed onto the carriers and the surface adsorption technique was successful. To obtain an even clearer picture, the PXRD, DSC and FESEM data were validated together. The results of PXRD studies are given in Figure 4c.

### 3.9. Microscopic Study

From the FESEM images, it was noticed that CUN possessed distinct edges, elongated formation and extended cuboidal-shaped structures indicating its crystallinity. On the other hand, GIQ9 exhibited irregular spherical, granular, unevenly surfaced structures, suggesting the amorphous nature of the sample. Further, AER-200 showed porous, spongy, permeable, poriferous-shaped structures indicating its highly amorphous nature. The FESEM images of CUN-S-SNEDDS revealed that the final formulation had dense, flaky, irregular structures which exhibited the amorphous nature of the formulation. This indicated that CUN in its isotropic form was successfully adsorbed onto the carriers, corresponding to better flow properties of the formulation after the usage of the surface adsorption technique [68]. The TEM images of CUN-L-SNEDDS and CUN-S-SNEDDS (reconstituted in water) were used to examine the morphological properties and droplet size of these formulations. The droplets were determined to be spherical and consistent in diameter. There was no evidence of droplet accumulation. According to the dynamic light scattering studies, CUN-L-SNEDDS showed a droplet size of 77.23 nm, and the droplet size of CUN-S-SNEDDS was 97.54 nm. In the TEM images, it was noticed that the developed CUN-L-SNEDDS and CUN-S-SNEDDS had a size range below 100 nm. This validated the previous reports of the droplet size. The results of the morphological studies are given in Figure 5.

### 3.10. In Vitro Drug Release Studies

From the dissolution study, it was inferred that CUN-L-SNEDDS and CUN-S-SNEDDS showed nearly 90% release of CUN in the first 5 min, indicating an enhancement in CUN’s solubility and excellent adsorption of the drug onto the carriers GIQ9 and AER-200. However, it was noticed that only 18.8% of naïve CUN was dissolved in 60 min. This indicated a limited rate of dissolution of CUN owing to its lipophilic nature [69]. However, the CUN that was loaded in the isotropic mixture in a completely soluble form was self-emulsified upon reaching the buffered aqueous medium, leading to complete solubility of CUN in the dissolution medium. There was a 5.1-fold increase in the dissolution rate of CUN loaded in SNEDDSs as compared to naïve CUN. The increase in the dissolution rate of CUN was found to be superior to other CUN-SNEDDS-based formulations reported in the literature, which showed 2.5-fold [70] and 3.8-fold [64] increases. To compare the release profile of naïve CUN, CUN-L-SNEDDS and CUN-S-SNEDDS, f2 analysis and one-way ANOVA were carried out. It was found that the f2 value was less than 50 and the *p* value was less than 0.0001 when comparing the naïve CUN and the SNEDDS formulation. This suggested that the dissolution rate and solubility of the CUN were enhanced after formulation into SNEDDSs. The *p* value when comparing the CUN-L-SNEDDS and CUN-S-SNEDDS was 0.96, and the f2 value was 94, indicating similar release profiles. This indicated that there was no alteration in the release of CUN from SNEDDSs after solidification using GIQ9 and AER-200 as carriers and the usage of the surface adsorption technique [71]. The results of in vitro studies are given in Figure 6a.

### 3.11. Pharmacokinetic Study

The naïve CUN and CUN-S-SNEDDS were orally administered to rats of two different groups. Noncompartmental analysis was carried out and the pharmacokinetic parameters were calculated. It is pertinent to add here that the rate of absorption was determined by the T_max_, and the extent of absorption was determined by AUC. Moreover, the C_max_ indicates both [47]. The findings indicated that the T_max_ of the naïve drug occurred at 0.5 h. For CUN-S-SNEDDS, it was at 1.5 h. The C_max_ for naïve CUN and CUN-S-SNEDDS was 42 ng/mL and 532 ng/mL, respectively, which indicated the increased absorption of the drug after the conversion into the S-SNEDDS and improvement in solubility using GIQ9 as well as AER-200 as carriers. The t_1/2_ was 2 h for raw CUN and 9 h for CUN-S-SNEDDS. A larger AUC_0- ∞_ for CUN-S-SNEDDS was indicative of the fact that CUN bioavailability was increased after formulation into S-SNEDDSs [49]. The bioavailability of the developed CUN-S-SNEDDS was increased by 61.7-fold. The increase in bioavailability of CUN was found to be superior to other CUN-SNEDDS-based formulations reported in the literature which showed 6-fold [72], 62-fold [73] and 10.77-fold [74] increases. Further, the mean residence time of naïve CUN was increased from 2.95 h to 15.63 h, indicating the improved therapeutic potential of the drug and increased circulation. Overall, the pharmacokinetic study indicated that upon solidification, the solubility and bioavailability of the poorly soluble drug CUN were enhanced, especially after its usage as a carrier. The results of the pharmacokinetic study are summarized in Table 3 and Figure 6b.

### 3.12. In Vitro/In Vivo Correlations (IVIVCs)

Like other BCS class II drugs, CUN also exhibits a dissolution rate-limited absorption as well as a point-to-point correlation between in vitro dissolution studies and in vivo pharmacokinetic activity [1]. The results revealed a higher degree of nonlinear fitting between the parameters of the drug dissolved and drug absorbed. For CUN and for CUN-S-SNEDDS, it was R^2^ = 0.966 and R^2^ = 0.907, respectively. The quadratic model indicated superior data fitting for the curved model. The CUN-S-SNEDDS formulation demonstrated nonlinear fitting; this was attributed to the rapid release of the drug as compared to the absorption rate [75], as seen in the case of other self-emulsifying drug delivery systems. The results of the pharmacokinetic study are given in Figure 6c.

## 4. Conclusions

In the present study, an attempt has been made to develop GIQ9 as an effective carrier for S-SNEDDS formulation which would help enhance the mechanical strength, dissolution rate and bioavailability of CUN. The selection of carriers was performed using multivariate data analysis, and optimization was performed using central composite design. The use of GIQ9 proved to be an effective carrier for S-SNEDDSs as it showed excellent entrapment efficiency, zeta potential and micromeritic properties. The similar release profiles between CUN-L-SNEDDSs and CUN-S-SNEDDSs may indicate that solidification using GIQ9 and AER-200 did not impact the performance of drug release. Further, they also revealed the use of GIQ9 as an excellent carrier for lipophilic-based drug delivery systems. Owing to the positive outcomes of the study, the same approach can be used to develop S-SNEDDSs of other lipophilic drugs.

## Figures and Tables

**Figure 1 pharmaceutics-14-02395-f001:**
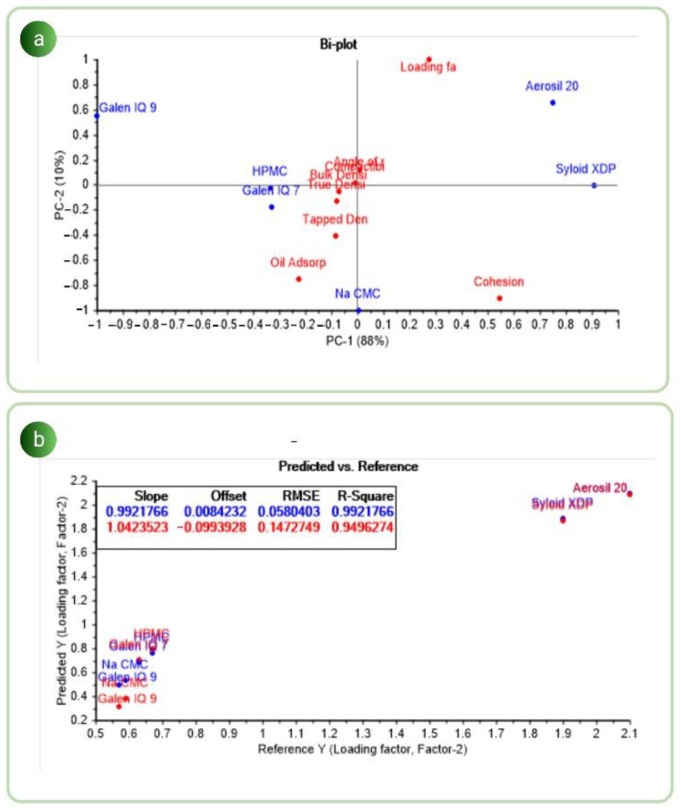
(**a**) Biplot of PC1 (88%) vs. PC2 (10%); (**b**) PLS correlation plot between predicted and reference loading factor (Lf).

**Figure 2 pharmaceutics-14-02395-f002:**
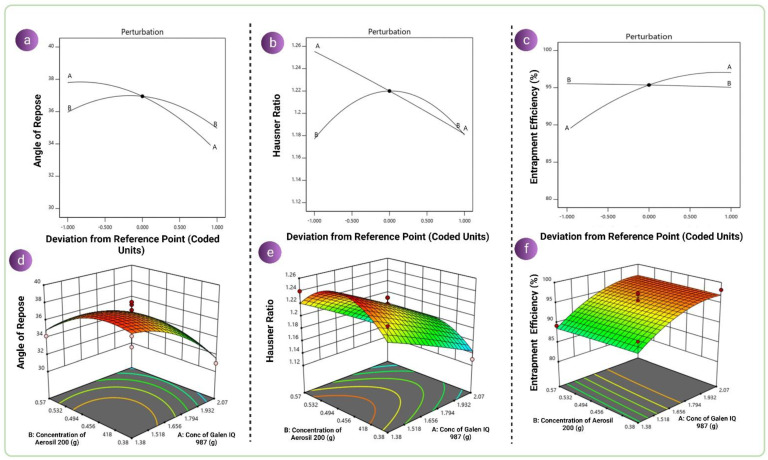
Perturbation plots for (**a**) AR, (**b**) HR amd (**c**) EE and 3D contours of (**d**) AR, (**e**) HR and (**f**) EE.

**Figure 3 pharmaceutics-14-02395-f003:**
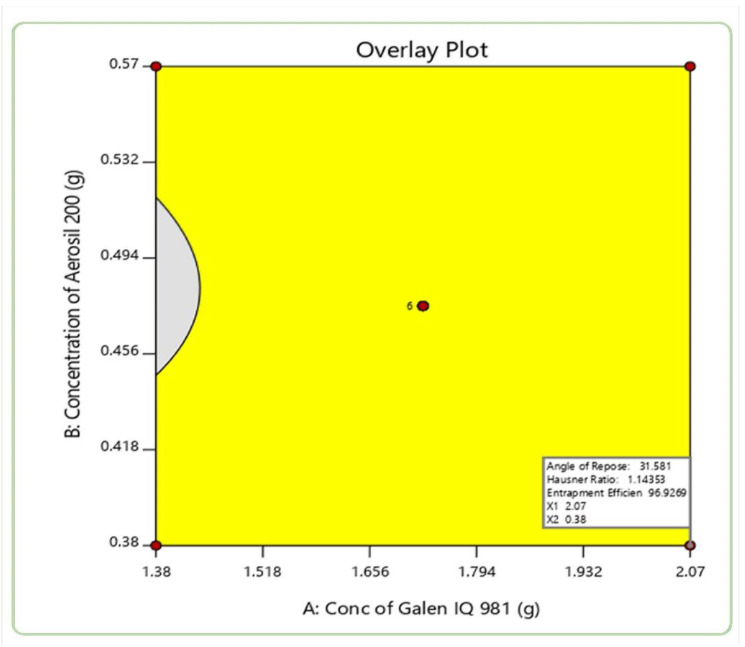
Graphical optimized plot from CCD.

**Figure 4 pharmaceutics-14-02395-f004:**
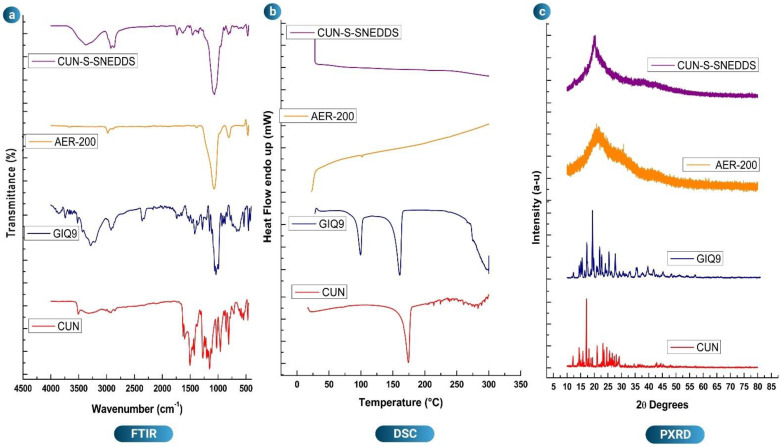
Overlay plots of (**a**) FTIR, (**b**) DSC and (**c**) PXRD of naïve CUN, excipients and optimized formulation.

**Figure 5 pharmaceutics-14-02395-f005:**
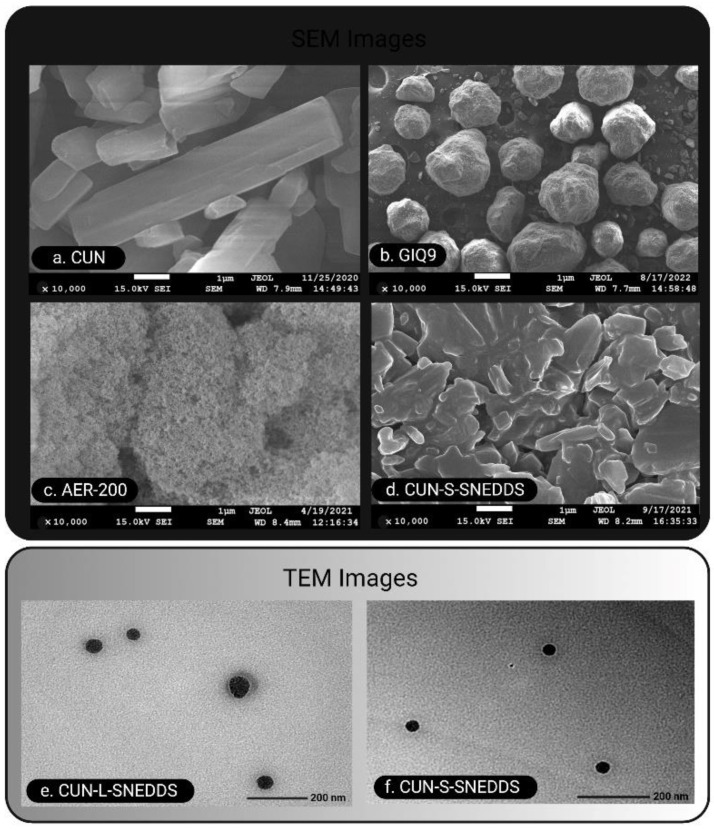
FESEM images of (**a**) CUN, (**b**) GIQ9, (**c**) AER-200 and (**d**) CUN-S-SNEDDS; TEM images of (**e**) CUN-L-SNEDDS and (**f**) CUN-S-SNEDDS.

**Figure 6 pharmaceutics-14-02395-f006:**
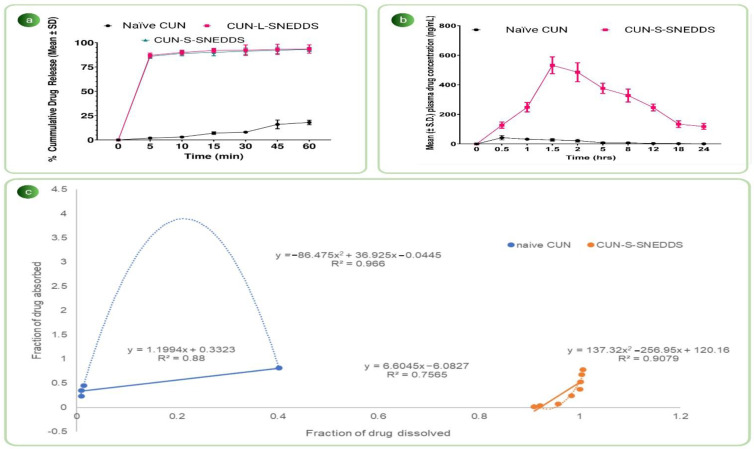
Results of (**a**) in vitro dissolution studies; (**b**) in vivo pharmacokinetic studies; (**c**) IVIVCs of naïve CUN and CUN-S-SNEDDS.

**Table 1 pharmaceutics-14-02395-t001:** Micromeritic properties and Lf values of various carriers used for solidification (number of replicates *n* = 3).

Carrier	AR (°)	BD (g/cm^3^)	TAD (g/cm^3^)	TRD (g/cm^3^)	Compatibility	Cohesion	Oil Adsorption Capacity (g)	Lf	HR	CI (%)
GIQ9	34.84 ± 0.04	0.66 ± 0.25	0.76 ± 0.13	0.72 ± 0.15	0.10 ± 0.10	0.63 ± 0.34	1.73 ± 0.13	0.57 ± 0.21	1.15 ± 0.32	13.16 ± 0.42
GIQ7	36.90 ± 0.74	0.33 ± 0.13	0.46 ± 0.14	0.39 ± 0.56	0.05 ± 0.14	2.15 ± 0.54	1.58 ± 0.24	0.63 ± 0.45	1.39 ± 0.45	28.26 ± 0.54
SXDP	23.52 ± 0.09	0.26 ± 0.09	0.34 ± 0.37	0.29 ± 0.11	0.03 ± 0.23	10.24 ± 0.05	0.53 ± 0.32	1.90 ± 0.34	1.31 ± 0.58	23.55 ± 0.21
AER-200	41.78 ± 1.16	0.04 ± 0.03	0.06 ± 0.02	0.05 ± 0.42	0.05 ± 0.18	3.61 ± 0.13	0.48 ± 0.03	2.10 ± 0.54	1.50 ± 0.51	33.33 ± 0.43
HPMC	43.10 ± 1.32	0.32 ± 0.23	0.55 ± 0.07	0.38 ± 1.04	0.06 ± 0.04	2.05 ± 0.45	1.47 ± 0.42	0.67 ± 0.34	1.72 ± 0.23	41.82 ± 0.11
Na CMC	50.54 ± 0.36	0.37 ± 0.07	0.76 ± 0.04	0.47 ± 0.19	0.07 ± 0.04	47.95 ± 0.06	1.67 ± 0.23	0.59 ± 0.17	2.05 ± 0.07	51.32 ± 0.44

**Table 2 pharmaceutics-14-02395-t002:** Central composite design-based factor level and responses (number of replicates *n* = 3).

	Factor 1	Factor 2	Response 1	Response 2	Response 3
Run	A: Conc of Galen IQ 981 (g)	B: Conc of Aerosil 200 (g)	Angle of Repose (°)	Hausner Ratio	Entrapment Efficiency (%)
1	2.07	0.38	31.2 ± 0.23	1.13 ± 0.43	98.2 ± 0.13
2	2.07	0.57	32.6 ± 0.47	1.12 ± 0.76	96.3 ± 0.42
3	1.73	0.48	38.1 ± 0.87	1.22 ± 0.56	97.1 ± 0.34
4	1.73	0.48	37.2 ± 0.34	1.23 ± 0.18	95.6 ± 0.53
5	1.73	0.48	37.8 ± 0.43	1.22 ± 0.61	94.3 ± 0.34
6	1.24	0.48	38.4 ± 0.08	1.25 ± 0.89	83.4 ± 0.54
7	1.38	0.38	36.4 ± 0.34	1.23 ± 0.31	92.3 ± 0.17
8	1.38	0.57	34.2 ± 0.48	1.24 ± 0.57	89.3 ± 0.42
9	2.21	0.48	31.4 ± 0.54	1.18 ± 0.28	96.4 ± 1.04
10	1.73	0.48	38.1 ± 0.63	1.23 ± 0.07	97.4 ± 0.56
11	1.73	0.48	36.3 ± 0.27	1.19 ± 0.23	93.2 ± 0.34
12	1.73	0.48	34.2 ± 0.43	1.23 ± 0.45	94.6 ± 0.45
13	1.73	0.61	33.4 ± 0.32	1.14 ± 0.56	95.3 ± 0.88
14	1.73	0.34	35.8 ± 1.22	1.13 ± 0.83	93.2 ± 0.23

**Table 3 pharmaceutics-14-02395-t003:** Summary of pharmacokinetic study, (number of replicates (*n*) = 6).

Parameter	Unit	Naïve CUN	CUN-S-SNEDDS
t_1/2_	h	2.00 ± 0.23	9.95 ± 0.31
Tmax	h	0.50 ± 0.32	1.50 ± 0.24
Cmax	ng/ml	42.00 ± 0.18	532.00 ± 0.06
AUC _0-t_	ng/mL·h	101.00 ± 0.13	5966.25 ± 0.22
AUC_0-∞_	ng/mL·h	124.14 ± 0.21	7660.88 ± 0.34
MRT_0-∞_	h	2.95 ± 0.03	15.64 ± 0.04

## Data Availability

Not applicable.

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
