# Peer review of "Multivariate Data Analysis and Central Composite Design-Oriented Optimization of Solid Carriers for Formulation of Curcumin-Loaded Solid SNEDDS: Dissolution and Bioavailability Assessment"

_pharmaceutics, 2022, doi:10.3390/pharmaceutics14112395_

Round 1

Reviewer 1 Report

The manuscript “Quality by design-oriented development of isomalt-aerosil based liquisolid SNEDDS powder loaded with curcumin: formulation, optimization and bioavailability assessment” seems interesting and looks good. Thus, I recommend considering the following points to improve the quality of the research.

Comments and suggestions for authors

Table 1, 2, and 3: Each experiment should be carried out in triplicate (minimum n =3 or greater). I wonder do the authors carried these experiments in triplicate. I could not see an indication in the table or the table legends. Also, indicate each data with standard deviation (SD) in each table. Also, in figures where applicable.

Abstract; Line 48: Please correct this sentence (the word PDI is missing and need to switch words as well, i.e., switch particle size and zeta potential).

Line 211: Indicate drug loading with SD. Here, drug loading means entrapment efficiency, thus, it is better to write entrapment efficiency in place of drug loading. Readers might be confused with the word drug loading capacity.

Line 296: Pharmacokinetic study: S-SNEDDS is solid, might be powder, how authors have administered it to the rats? Do authors use any vehicle or capsule for administering S-SNEDDS? Please indicate in the manuscript.

Line 300: CUN-CUN-S-SNEDDS; CUN is repeated, please correct it.

Reviewer 2 Report

The manuscript “Quality by design-oriented development of isomalt-aerosil based liquisolid SNEDDS powder loaded with curcumin: formulation, optimization and bioavailability assessment” by Leander Corrie et al. would like to describe the formulation of an S-SNEDDS system to deliver curcumin by increasing its pharmacokinetics. However, there is some discordance between what is reported in the abstract and introduction versus what is described in the results. Even the title, in my opinion, does not represent the results presented. In fact, taking into consideration the title, abstract and introduction, the results presented in sections 3.1 to 3.4 are useless. In order to make the best use of the study carried out for screening of different carriers and multivariance analyses, the title should be changed completely, and the overall layout of the article, which in any case is quite confusing and unclear.

For example, section 3.1 lacks discussion, and section 3.2 refers to Figure 2.c when Figure 1 has not yet been presented. The confusion of this first part of presenting the results is, in my opinion, accentuated by the fact that never is it mentioned what combination is chosen and why.

 Other major points are:

1. the figure legends should be more explanatory;

2. check abbreviations of compounds (e.g., in Table 1, there are full names of compounds when acronyms should be included; in Figure 3 c there is A-200 and not AER-200);

3. the SEM images should all be at the same magnification;

4. there is a lack of comparison between the system reported here and what is already in the literature;

5. check English and typos.

Reviewer 3 Report

It is a technical paper, I have not a lot of experience in this style of manuscripts

Round 2

Reviewer 1 Report

I believe that the manuscript has better now. However, I found still there are some serious issues/errors:

1) In line 327 (PK study), the authors have divided animal into three groups? Why the authors have only presented two groups in the data (Figure 6 b and table 3). If three groups, update the data accordingly.

2) Table 3: Please correct the number of replicates. (In line 327, each group consists of six animals).

3) in line 450, Please correct the drug loading, make it consistent all over the manuscript. 

Reviewer 2 Report

the authors addressed most of my concerns.

Two more issue have to be corrected:

lane 327: the animals have been divided in two groups, so please correct the statement "The animals were divided into three groups, each with six animals";

lane 677: n should be 6, if each group contain 6 animals.

Round 3

Reviewer 1 Report

The authors have corrected the manuscript as per my comments and suggestions, except one comment.

However, for my last comment, authors had misunderstood me. I just asked authors to change the word drug loading present in the equation of line 446-450 to entrapment efficiency so as to make consistency in the manuscript. However, authors have entirely changed the word 'entrapment efficiency' to the word 'drug loading'.

???? ??????? (?3) = +95.37 + 3.91? − 0.2413? + 0.2750?? − 2.25?2 − 0.0708?2.

Thus, please keep entrapment efficiency as it is in the manuscript for easy understanding of the readers, and just change the word 'drug loading' to 'entrapment efficiency' in the above equation.
